# Whole-Genome Sequencing Analysis of a *stx*-Negative *Escherichia* *coli* O63:H6 Isolate Associated with Hemolytic Uremic Syndrome

**DOI:** 10.3390/diagnostics11101823

**Published:** 2021-10-02

**Authors:** Tae Yeul Kim, Tae-Min La, Taesoo Kim, Sun Ae Yun, Sang-Won Lee, Hee Jae Huh, Nam Yong Lee

**Affiliations:** 1Department of Laboratory Medicine and Genetics, Samsung Medical Center, Sungkyunkwan University School of Medicine, Seoul 06351, Korea; voltaire0925@gmail.com (T.Y.K.); micro.lee@samsung.com (N.Y.L.); 2College of Veterinary Medicine, Konkuk University, Seoul 05029, Korea; white_tm@naver.com (T.-M.L.); taesoo95811@naver.com (T.K.); 3Center for Clinical Medicine, Samsung Biomedical Research Institute, Samsung Medical Center, Seoul 06351, Korea; sm10040730@naver.com

**Keywords:** *E. coli* O63:H6, STEC, EHEC, *stx* loss, hemolytic uremic syndrome, whole-genome sequencing

## Abstract

Shiga toxin-encoding genes (*stx*) of enterohemorrhagic *Escherichia* *coli* (EHEC) can be lost during infection or in vitro cultivation, and in clinical practice, it is difficult to distinguish EHEC that have lost *stx* (EHEC-LST) from enteropathogenic *E*. *coli* (EPEC), as both are *stx*-negative and *eae*-positive. In this study, we performed whole-genome sequencing (WGS) of a *stx*-negative, *eae*-positive *E*. *coli* O63:H6 isolate from a child with hemolytic uremic syndrome and compared its genome with those of nine *E*. *coli* O63:H6 strains in public databases. Virulence gene profiles were analyzed and core-genome multilocus sequence typing (cgMLST) was conducted. The virulence gene profile of our isolate was consistent with EHEC, except for the absence of *stx*, and the isolate clustered with seven EHEC strains but was distant from two EPEC strains in cgMLST. In *genome alignment*, our isolate exhibited a high nucleotide identity with EHEC strain 377323_2f but displayed a gap corresponding to the *stx*-harboring prophage sequence. Overall, our isolate was genetically closely related to EHEC strains, consistent with this being an EHEC-LST strain. As EHEC-LST may be misdiagnosed as EPEC in routine laboratories, comparative genomic analysis using WGS can be useful to determine whether *stx*-negative and *eae*-positive isolates are EHEC-LST or EPEC.

## 1. Introduction

Shiga toxin-producing *Escherichia coli* (STEC) can cause human illnesses ranging from uncomplicated diarrhea to life-threatening complications such as hemolytic uremic syndrome (HUS). More than 200 STEC serotypes are known to be associated with human illnesses, among which O157:H7 is considered the most common serotype [1]; however, in recent years, sporadic cases and outbreaks caused by non-O157 STEC serotypes have been increasingly reported [1]. The serogroup/serotype distribution of non-O157 STEC strains varies from country to country. In the United States, six serogroups (O26, O103, O111, O121, O45, and O145) account for the majority of non-O157 STEC cases [2]. In Germany, the Netherlands, and Belgium, a substantial proportion of non-O157 STEC strains belong to the serotype O63:H6 [3,4,5,6], but this serotype has been rarely reported elsewhere.

Shiga toxin (Stx) is the primary virulence factor of STEC involved in the pathogenesis of HUS, and the genes encoding Stx (*stx*) are located in bacteriophage genomes integrated into the bacterial chromosome [7]. In addition to *stx*, a subset of STEC strains, termed enterohemorrhagic *E*. *coli* (EHEC), possess the locus of enterocyte effacement (LEE) pathogenicity island containing *eae*, which encodes intimin, a protein required for intimate bacterial attachment to the host intestinal epithelium [7]. Similar to EHEC, enteropathogenic *E*. *coli* (EPEC) carries the LEE pathogenicity island, but in contrast to EHEC, EPEC lacks *stx* and is not associated with HUS [7,8]. EPEC strains are divided into typical EPEC and atypical EPEC based on the presence of the EPEC adherence factor plasmid carrying the bundle-forming pilus (*bfp*) operon, a cluster of 14 genes that includes *bfpA*; typical EPEC strains are *stx*-negative, *eae*-positive, and *bfp*-positive, whereas atypical EPEC strains are *stx*-negative, *eae*-positive, and *bfp*-negative [7,9]. However, EHEC strains can lose *stx*-harboring prophages during infection or in vitro cultivation [10,11,12], and in clinical practice, it is difficult to distinguish EHEC that have lost *stx* (EHEC-LST) from EPEC, as both are *stx*-negative and *eae*-positive.

Recently, we obtained a *stx*-negative, *eae*-positive *E*. *coli* O63:H6 isolate from a 3-year-old child diagnosed with HUS based on the presence of non-immune hemolytic anemia, thrombocytopenia, and acute kidney injury and performed whole-genome sequencing (WGS) to determine whether this isolate was EHEC-LST or EPEC.

## 2. Materials and Methods

A stool sample was collected 7 days after the onset of gastrointestinal symptoms (abdominal pain and vomiting) and sent for culture. Species identification was carried out using the Vitek 2 system (bioMérieux, Marcy l’Etoile, France). Serotyping was performed by agglutination test using O and H antisera (Denka Seiken, Tokyo, Japan). The presence of *stx1*, *stx2*, *stx2f*, and *eae* in both stool sample and *E*. *coli* isolate was tested by PCR using the primers described in Appendix A.

We performed WGS for the molecular characterization of the isolate. Genomic DNA was extracted using the MagAttract HMW DNA kit (Qiagen, Hilden, Germany). To obtain the complete genome of our isolate, long read sequencing was performed using MinION (Oxford Nanopore Technologies, Oxford, UK). To overcome the high error rate of Nanopore long read sequencing, short read sequencing was also performed using HiSeq X Ten (Illumina, San Diego, CA, USA), and the de novo hybrid assembly of Illumina short reads and Nanopore long reads was performed using Unicycler v0.4.7 [13]. The assembled genome was annotated and visualized using Geneious Prime (Biomatters, Auckland, New Zealand) [14], and then the sequence was compared with the genome sequences of nine *E*. *coli* O63:H6 strains found in public databases: GenBank 377323_2f; EnteroBase 179620, 77AZUC, 8X4S73, SSI-AC286, SSI-AC307, OLC2140, 190561, and 191861. We identified virulence genes using VirulenceFinder 2.0 [15] and performed in silico multilocus sequence typing (MLST) using MLST 2.0 [16]. Core-genome MLST (cgMLST) was performed using the EnteroBase scheme [17], and a minimum spanning tree was constructed using GrapeTree [18]. Pairwise genome alignment was performed using Mauve v2.4.0 [19] and BLAST Ring Image Generator (BRIG) [20].

## 3. Results

Colorless colonies grew on a sorbitol-MacConkey agar plate, and the isolate (strain EC12) was identified as *E*. *coli* O63:H6. Both the stool sample and strain EC12 isolated from the sample tested negative for *stx1*, *stx2*, and *stx2f* but positive for *eae*. Illumina HiSeq X Ten generated a total of 13,227,764 reads, yielding 403× coverage of the genome. Nanopore MinION generated 324,092 reads, with a mean read length of 6922 bases and a read length N50 of 23,635, resulting in 467× coverage of the genome. The genome sequence of strain EC12 was assembled into three circular contigs: the 4,951,794 bp chromosome; a 163,194 bp IncFIB/FII-type plasmid named pEC12_1; and a 58,414 bp IncI2-type plasmid named pEC12_2 (Figure 1). The chromosome was predicted to contain 4752 protein-coding genes, 22 rRNA genes, and 96 tRNA genes. In the pEC12_1, 201 protein-coding genes and one tRNA gene were predicted. In the pEC12_2, 76 protein-coding genes were predicted. Consistent with the PCR results indicating that it was *stx*-negative but *eae*-positive, the chromosome of strain EC12 contained the LEE pathogenicity island but lacked *stx*-harboring prophages. Plasmid pEC12_1 contained the *bfp* operon. The virulence genes identified in the genome of strain EC12 and other *E*. *coli* O63:H6 strains are compared in Table 1. Seven *stx*-positive, *eae*-positive strains (377323_2f, 179620, 77AZUC, 8X4S73, SSI_AC286, SSI_AC307, and OLC2140) and two *stx*-negative, *eae*-positive, *bfp*-negative strains (190561 and 191861) were classified as EHEC and atypical EPEC, respectively. Strain EC12 had the virulence gene profile consistent with EHEC, except for *stx*. In silico MLST showed that strain EC12 and EHEC strains belong to sequence type 583 (ST583), whereas atypical EPEC strains belong to ST122. The minimum spanning tree based on cgMLST results revealed that strain EC12 was closely related to EHEC strains but distinct from atypical EPEC strains (Figure 2). In genome alignment using Mauve and BRIG, strain EC12 exhibited a high nucleotide identity with EHEC strain 377323_2f but displayed a gap of around 42.5 kbp corresponding to the *stx*-harboring prophage sequence (Figure 3A,B). Furthermore, plasmid pEC12_1 contained the whole sequence of p377323_2f, a plasmid of strain 377323_2f, in Mauve alignment (Figure 3C). The complete genome sequences of strain EC12 have been deposited in GenBank under accession no. CP070229 (chromosome), CP070230 (pEC12_1), and CP070231 (pEC12_2).

## 4. Discussion

So far, *stx2f* has been the only *stx* subtype found in *E*. *coli* O63:H6 strains [3,4,5,6]. STEC strains carrying *stx2f* are generally associated with mild disease, potentially because these strains carry a relatively small number of virulence genes [5]. However, *stx2f*-positive STEC strains can cause severe disease such as HUS, albeit rarely [6,21]. Although *stx2f*-positive STEC strains are frequently found in pigeons, STEC serotypes associated with human diseases, such as O63:H6, are not found in pigeons, suggesting that the *stx2f*-positive STEC strains responsible for human diseases do not originate from the pigeon reservoir [22].

Infections caused by *stx2f*-positive STEC strains tend to be underreported, as many molecular assays for laboratory diagnosis of STEC infection are not designed to detect *stx2f* [5,6]. The addition of *stx2f* primers to existing PCR assays can be useful for accurate diagnosis of STEC infections, particularly in countries where *stx2f*-carrying STEC strains are frequently encountered.

Of particular note, strain EC12 was positive for *eae* but negative for *stx*. Strain EC12 was initially suspected to be an EHEC-LST strain as it was isolated from an HUS patient. Additionally, as the stool sample of this patient, which was collected 7 days after the onset of gastrointestinal symptoms, was confirmed by PCR to be positive for *eae* but negative for *stx*, strain EC12 is presumed to have lost *stx* in vivo during infection rather than during in vitro culture. The loss of *stx* is a serotype-related phenomenon and has been found in serotypes O26:H11/NM (nonmotile), O103:H2/NM, O145:H28/NM, and O157:H7/NM [10,11,12]. This study provides the first evidence that *stx* loss can occur in serotype O63:H6.

When using PCR-based assays targeting only *stx* and *eae*, EPEC strains isolated from HUS patients are at risk of being misclassified as EHEC-LST. Whole genome comparison is very useful to determine whether *stx*-negative, *eae*-positive *E*. *coli* strains isolated from HUS patients are EHEC-LST. A recent WGS study reported that *stx*-negative, *eae*-positive *E*. *coli* O157:H7/NM strains clustered together with EHEC strains but not with EPEC strains in cgMLST [11]. Furthermore, virulence gene analysis using WGS data revealed that these strains shared a similar virulence gene profile with EHEC strains rather than EPEC strains [11]. Another WGS study demonstrated that a *stx*-negative, *eae*-positive *E*. *coli* O145:H28 strain had the identical cgMLST and non-*stx* virulence gene profiles with a *stx*-positive, *eae*-positive *E*. *coli* O145:H28 strain isolated from the same patient [12]. In pairwise genome alignment, the *stx*-negative, *eae*-positive strain was shown to lack the 9 kbp region corresponding to the partial sequence of the *stx*-harboring prophage [12]. In the present study, phylogenetic analysis using cgMLST showed that strain EC12 clustered together with EHEC strains but not with EPEC strains. Furthermore, this strain shared non-*stx* virulence genes with EHEC strains and exhibited a very high nucleotide identity with EHEC strain 377323_2f except for the *stx*-harboring prophage sequence. Taken together, our results indicate that strain EC12 is EHEC-LST, not EPEC. Interestingly, strain EC12 and EHEC strains were *nleB*-negative and *nleC*-positive, whereas EPEC strains were *nleB*-positive and *nleC*-negative. This finding suggests that, in serotype O63:H6, *nleB* and *nleC* may be good molecular markers for differentiating EHEC-LST from EPEC.

In summary, we isolated a *stx*-negative, *eae*-positive *E*. *coli* O63:H6 strain from a child with HUS. Comparative genomic analysis using WGS data revealed that this strain is genetically more closely related to EHEC than to EPEC, consistent with it being an EHEC-LST strain. Of diagnostic interest, EHEC-LST may be misdiagnosed as EPEC in routine laboratories, and comparative genomic analysis using WGS data can be useful to determine if *stx*-negative and *eae*-positive isolates are EHEC-LST or EPEC.

## Figures and Tables

**Figure 1 diagnostics-11-01823-f001:**
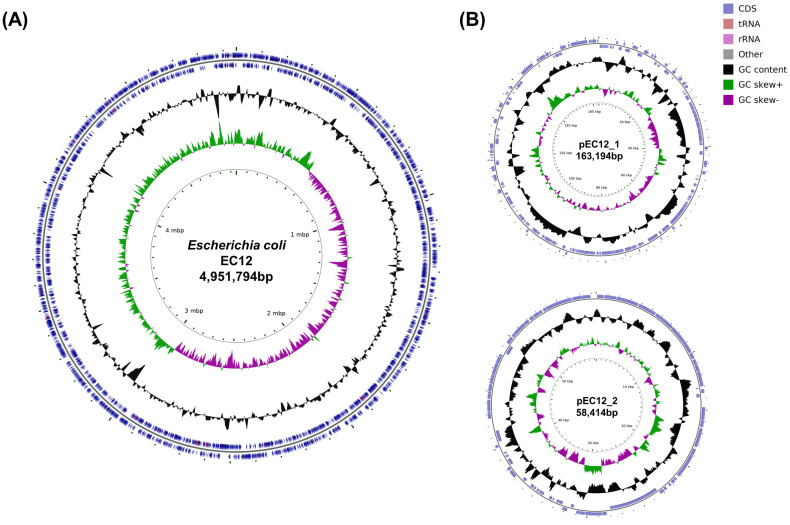
Circular maps of the chromosome (**A**) and two plasmids (**B**) of strain EC12. From the outer circle moving inward, circles 1 and 2 represent protein-coding genes (blue), tRNA genes (reddish brown), and rRNA genes (light purple) on the positive and negative strands; circle 3 shows the GC content (black); circle 4 shows the GC skew (positive GC skew, green; negative GC skew, purple); and circle 5 displays the genomic coordinates.

**Figure 2 diagnostics-11-01823-f002:**
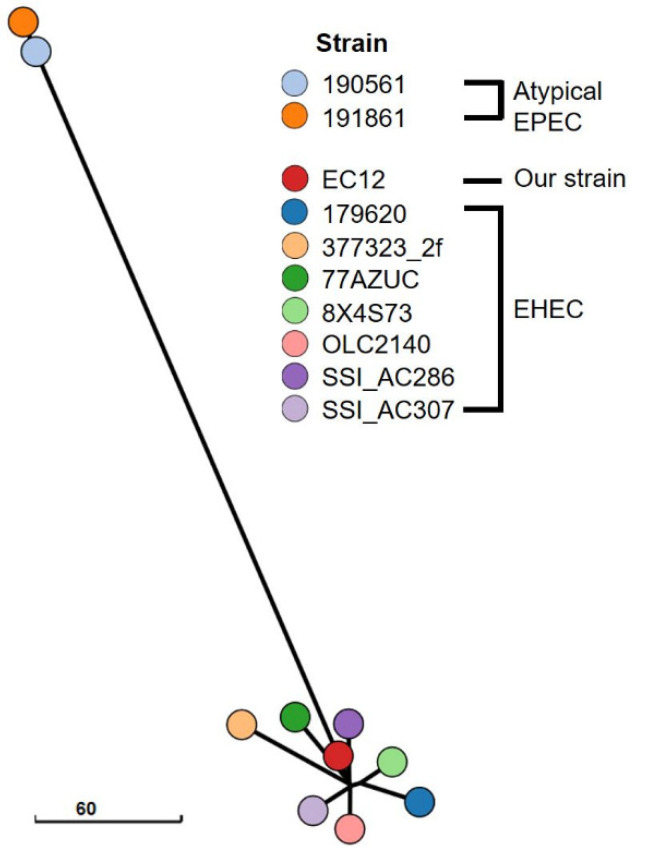
Minimum spanning tree based on cgMLST profiles of 10 *E*. *coli* O63:H6 strains included in this study. The branch lengths reflect the number of allelic differences between strains.

**Figure 3 diagnostics-11-01823-f003:**
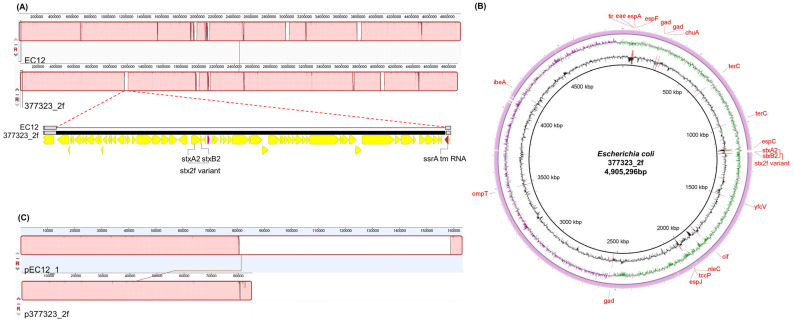
Genome comparison of strains EC12 and 377323_2f. (**A**,**B**) Chromosome comparisons using Mauve (**A**) and BRIG (**B**) demonstrate a gap of around 42.5 kbp in EC12 corresponding to the *stx*-harboring prophage sequence. (**C**) Plasmid alignment using Mauve shows that pEC12_1 contains the whole sequence of p377323_2f. In Mauve, identically colored boxes represent homologous regions shared by the aligned genomes.

**Table 1 diagnostics-11-01823-t001:** Distribution of virulence genes identified in the genomes of *E*. *coli* O63:H6 strains.

Strain	*astA*	*bfpA*	*chuA*	*cif*	*eae*	*espA*	*espC*	*espF*	*espJ*	*gad*	*ibeA*	*nleB*	*nleC*	*ompT*	*stx*	*tccP*	*terC*	*tir*	*yfcV*
EC12	+	+	+	+	+	+	+	+	+	+	+	−	+	+	−	+	+	+	+
377323_2f	+	+	+	+	+	+	+	+	+	+	+	−	+	+	+ *	+	+	+	+
179620	+	+	+	+	+	+	+	+	+	+	+	−	+	+	+ *	−	+	+	+
77AZUC	+	+	+	+	+	+	+	+	+	−	+	−	+	+	+ *	−	+	+	+
8X4S73	+	+	+	+	+	+	+	+	+	+	+	−	+	+	+ *	+	+	+	+
SSI_AC286	+	+	+	+	+	+	+	+	+	+	+	−	+	+	+ *	−	+	+	+
SSI_AC307	+	+	+	+	+	+	+	+	+	−	+	−	+	+	+ *	−	+	+	+
OLC2140	−	−	+	+	+	+	+	+	+	−	+	−	+	+	+ *	+	+	+	+
190561	−	−	+	+	+	+	+	+	+	+	+	+	−	+	−	−	+	+	+
191861	−	−	+	+	+	+	+	+	+	+	+	+	−	+	−	−	+	+	+

* The *stx* subtype was *stx2f*.

## Data Availability

The data presented in this study are available on request from the corresponding author.

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
