# Peer review of "Whole-Genome Sequencing Analysis of a stx-Negative Escherichia coli O63:H6 Isolate Associated with Hemolytic Uremic Syndrome"

_diagnostics, 2021, doi:10.3390/diagnostics11101823_

Round 1
Reviewer 1 Report
- What is the reason of using both Illumina 10X and Nanopore sequencing? The results do not indicate clearly the difference between short and long read sequencing. Can authors highlight advantage of using both techniques?
- Figures need formatting. Specifically, the alignment and presentation need to be improved for Fig 2.
- Sequencing data need to be discussed more in detail as WGS is the highlight of HUS.
Author Response
Reviewer #1
- What is the reason of using both Illumina 10X and Nanopore sequencing? The results do not indicate clearly the difference between short and long read sequencing. Can authors highlight advantage of using both techniques?
▶Authors’ response: We thank the reviewer for this constructive suggestion. We performed whole-genome sequencing using Oxford Nanopore MinION. A major advantage of Oxford Nanopore sequencing is the ability to produce ultra-long reads, allowing the complete genome assembly. In this study, obtaining the complete genome of our strain was necessary for comparative genomic analysis. However, Oxford Nanopore sequencing still has a high error rate (~10%), requiring hybrid assembly using highly accurate Illumina short reads. We have addressed this part in the Materials and Methods section as follows.
“To obtain the complete genome of our isolate, long read sequencing was performed using MinION (Oxford Nanopore Technologies, Oxford, UK). To overcome the high error rate of Nanopore long read sequencing, short read sequencing was also performed using HiSeq X Ten (Illumina, San Diego, CA, USA), and the de novo hybrid assembly of Illumina short reads and Nanopore long reads was performed using Unicycler v0.4.7 [13].” (line 89-94)
- Figures need formatting. Specifically, the alignment and presentation need to be improved for Fig 2.
▶Authors’ response: We thank the reviewer for this valuable suggestion. We have corrected Figure 2 to improve its alignment and presentation and to better describe that strain EC12 is closely related to EHEC strains but distinct from atypical EPEC strains. Moreover, we have revised Figures 1 and 3 by increasing the resolution of the images and changing the alignment of text, font size, and font type.
- Sequencing data need to be discussed more in detail as WGS is the highlight of HUS.
▶Authors’ response: Following the reviewer’s suggestion, we have added a brief summary of sequencing data in the Results section as follows (lines 107-110, 113-115). Furthermore, a detailed discussion of comparison with other WGS studies was added (lines 159-167).

Reviewer 2 Report
Dear Authors:
1- the abstract is too short and need to contain more info
2-discussion need to contain more details and more comparisions
3-the quality of fig 1 and 3 is low
Author Response
Reviewer #2
- The abstract is too short and need to contain more info.
▶Authors’ response: Following the reviewer’s suggestion, we have added more information in the abstract.
- Discussion need to contain more details and more comparisons.
▶Authors’ response: Following the reviewer’s suggestion, a detailed discussion of comparison with other studies was added as follows.
à “Whole genome comparison is very useful to determine whether stx-negative, eae-positive E. coli strains isolated from HUS patients are EHEC-LST. A recent WGS study reported that stx-negative, eae-positive E. coli O157:H7/NM strains clustered together with EHEC strains but not with EPEC strains in cgMLST [11]. Furthermore, virulence gene analysis using WGS data revealed that these strains shared a similar virulence gene profile with EHEC strains rather than EPEC strains [11]. Another WGS study demonstrated that a stx-negative, eae-positive E. coli O145:H28 strain had the identical cgMLST and non-stx virulence gene profiles with a stx-positive, eae-positive E. coli O145:H28 strain isolated from the same patient [12]. In pairwise genome alignment, the stx-negative, eae-positive strain was shown to lack the 9 kbp region corresponding to the partial sequence of the stx-harboring prophage [12].” (lines 159-167).
- The quality of fig 1 and 3 is low.
▶Authors’ response: We appreciate the reviewer’s valuable comment to help us to improve the quality of the manuscript. We have improved the quality of Figures 1 and 3 by increasing the resolution of the images and changing the alignment of text, font size, and font type.
Round 2
Reviewer 2 Report
Thank you very much for the revision